# Metal–Dielectric Polarization-Preserving Anisotropic Mirror for Chiral Optical Tamm State

**DOI:** 10.3390/nano12020234

**Published:** 2022-01-12

**Authors:** Natalya V. Rudakova, Rashid G. Bikbaev, Pavel S. Pankin, Stepan Ya. Vetrov, Ivan V. Timofeev, Kuo-Ping Chen, Wei Lee

**Affiliations:** 1Kirensky Institute of Physics, Federal Research Center KSC SB RAS, 660036 Krasnoyarsk, Russia; bikbaev@iph.krasn.ru (R.G.B.); p.s.pankin@mail.ru (P.S.P.); S.Vetrov@inbox.ru (S.Y.V.); tiv@iph.krasn.ru (I.V.T.); 2Siberian Federal University, 660041 Krasnoyarsk, Russia; 3Institute of Imaging and Biomedical Photonics, College of Photonics, National Yang Ming Chiao Tung University, 301 Sec. 2, Gaofa 3rd Road, Guiren Dist., Tainan 711010, Taiwan; kpchen@nctu.edu.tw (K.-P.C.); wei.lee@nycu.edu.tw (W.L.)

**Keywords:** chiral optical Tamm state, polarization-preserving anisotropic mirror, *Q*-factor, coupled mode theory

## Abstract

This numerical study demonstrates the possibility of exciting a chiral optical Tamm state localized at the interface between a cholesteric liquid crystal and a polarization-preserving anisotropic mirror conjugated to a metasurface. The difference of the proposed structure from a fully dielectric one is that the metasurface makes it possible to decrease the number of layers of a polarization-preserving anisotropic mirror by a factor of more than two at the retained *Q*-factor of the localized state. It is shown that the proposed structure can be used in a vertically emitting laser.

## 1. Introduction

Recently, a hot topic in photonics is the optimization of *Q*-factor for localized states. These can be conventional defect modes of a photonic crystal and optical Tamm states (OTSs) excited at the interface between two multilayer mirrors [1,2]. When one of the mirrors is replaced by a metallic film, the light wave appears bound to a plasmon, i.e., vibrations of free electrons at the metal surface. In this case, the localized state is called the Tamm plasmon polariton (TPP) [3], which is analogous to the electronic Tamm state in solid state physics [4]. The TPP manifests itself in experiments as a resonance in the optical spectra of a sample [5,6]. The interest in localized states of this type stems from their potential for use in various optical devices, including sensors [7,8,9], detectors [10,11,12], lasers [13,14,15], absorbers [16,17], filters [18,19,20], and solar cells [21,22].

The state obtained by using a chiral material as one of the mirrors is called the chiral optical Tamm state (COTS) [23]. The chiral medius differs from achiral one by the absence of the mirror symmetry of optical properties. A well-known example of a chiral medium is a cholesteric liquid crystal (CLC) [24,25]. A CLC consists of elongated molecules oriented in space in the form of a twisting helix. Due to the helical symmetry of the CLC permittivity tensor and the periodicity of a helical pitch, the CLC is a one-dimensional photonic crystal (PC) with a photonic band gap in its spectrum. Such a chiral structure leads to diffraction of light circularly polarized in the helical twisting direction. The light with the oppositely twisted circular polarization does not diffract and is transmitted without significant changes. Let us consider the transmission of light through a CLC and reflection of light from it. The reflected wave retains the circular polarization of the same sign. In this case, it is difficult to excite the OTS at the interface between the chiral and achiral mirrors. The difficulty lies in the fact that an achiral mirror reverses the polarization of the reflected light. Therefore, the wave diffracting in the chiral mirror turns into a non-diffracting wave and, after re-reflection, leaves the interface between the media [26].

There are several ways to eliminate the polarization loss for localizing the light at the interface with a chiral medium. The first way is to use a quarter-wave phase plate for the polarization correction [27]. The second way is to combine a chiral medium with an anisotropic mirror [23,28,29,30,31]. Such a mirror retains the polarization sign at the reflection of light, i.e., the wave reflected from the mirror boundary has the same polarization as the incident wave. The third way of eliminating the polarization loss is to use a multilayer polarization-preserving anisotropic mirror (PPAM) [32,33]. The advantage of the multilayer PPAM is the absence of absorption owing to fully dielectric anisotropic layers. To obtain a high-*Q* localized state at the CLC–PPAM interface, the PPAM must have a larger number of periods. However, such a structure is difficult to be implemented. In accordance with current technology requirements, one has to reduce the number of anisotropic layers until it reaches the experimentally available level described in Reference [28].

In this work, we attempted to combine the second and third ways and create a hybrid mirror consisting of a small number of dielectric layers and a metasurface (MS). The use of a hybrid mirror simplifies the fabrication of the structure and makes it possible to additionally optimize and control its spectral properties.

## 2. Description of the Model

First, let us consider the interface between a cholesteric and a PPAM (see the inset to Figure 1a). The parameters of the CLC are a step (period) of p=0.46 µm, a total CLC layer thickness of $L^{c}=4$ μm, and ordinary and extraordinary beam refractive indices of $n_{∥}=n_{e}^{c}=\sqrt{\varepsilon_{∥}}=\sqrt{\varepsilon_{e}^{c}}=1.45$ and $n_{⊥}=n_{o}^{c}=\sqrt{\varepsilon_{⊥}}=\sqrt{\varepsilon_{o}^{c}}=1.75$, respectively.

A multilayer PPAM consists of alternating identical uniaxial dielectric layers with different refractive indices of the ordinary ($n_{o}^{p}$) and extraordinary ($n_{e}^{p}$) beams and an arithmetic mean of the permittivities of ${\bar{\varepsilon }}^{p}=\left(\varepsilon_{o}^{p}+\varepsilon_{e}^{p}\right)/2$. The structure unit cell includes a pair of layers with the orthogonal (vertical and horizontal (V–H)) optical axes. The number of PPAM layers is $N^{p}$ and the structure period is 2a (*a* is the thickness of one layer). For certainty, we take a=117 nm; $n_{o}^{p}=\sqrt{\varepsilon_{o}^{p}}=1.45$ and $n_{e}^{p}=\sqrt{\varepsilon_{e}^{p}}=1.75$; the investigated PPAM consists of $N^{p}=20$ layers.

A numerical analysis of the spectral properties of the investigated structure and the field distribution at the resonant wavelength was made using the Berreman 4 × 4 transfer matrix method [34] and the finite-difference time-domain (FDTD) method. The nanostructures were illuminated from the CLC side by a plane wave normally incident along the *z* axis. In the FDTD calculation, the following parameters were used. Reflectance *R* was calculated at the top of the simulation box. The periodic boundary conditions were imposed at the lateral sides of the simulation box and the perfectly matched layer (PML) boundary conditions were adopted for the remaining top and bottom sides.

The reflection spectra of the CLC–PPAM structure calculated by the Berreman and FDTD methods show that, at the interface between the CLC and the PPAM, the COTS is excited [33], which manifests itself as a resonance in the band gap center (Figure 1a). The normalized local field intensity distribution at the COTS wavelength is presented in Figure 1b, which shows that the field is localized at the CLC–PPAM interface. Importantly, the resonance line position can be effectively tuned in frequency by varying angle $\varphi_{1}$ between the CLC and the PPAM optical axes. Due to the great number of layers in the PPAM, the COTS *Q*-factor attains Q≈1500. However, a PPAM with a large number of layers is difficult to be realized.

Below, we show that the COTS can be excited with a thinner PPAM by incorporating the initial structure with a MS experimentally studied in Reference [28]. A schematic of the proposed structure is presented in Figure 2a. The MS is an array of rectangular gold [35] nanobricks arranged on a 100-nm-thick SiO_2_ layer deposited onto a 200-nm-thick reflecting gold plate. The refractive index of SiO_2_ thin layer was taken to be 1.45. Nanobricks have a length of 190 nm, a width of 70 nm, and a height of 70 nm. All of them are oriented with their long side along the *y* axis. A schematic of the MS is presented in Figure 2a. To accurately reproduce the nanobrick shape for calculating the MS in the FDTD simulation, an adaptive mesh was used. In order to fulfill the one-dimensional assumption for the Berreman method, the subwavelength nanobrick array was reasonably considered homogeneous in x-y-directions at the wavelength scale. It was presented as an anisotropic layer with refractive indices $n_{o}^{ms}=0.5$ and $n_{e}^{ms}=5.5$, respectively.

## 3. Results and Discussion

Figure 2b,c shows comparative reflectance spectra of the proposed structure calculated by the Berreman and FDTD methods and the destribution of the local field intensity at the COTS wavelength respectively. As compared with Figure 1a, the number of PPAM layers was decreased from 20 to 8; the structure thickness, excluding the CLC layer and the substrate, decreased from 2.3 to 1.3 μm. It can be seen that, in this case, the COTS *Q*-factor remains comparable to the resonance *Q*-factor in a purely dielectric structure (see Figure 3). It is important to note that, for PPAM with the number of layers less than 12 without metasurface, a COTS is not spectrally resolved. Moreover, the attached metasurface provides the excitation of a localized state even with fewer layers of PPAM. Thus, when $N^{p}=0$ (cholesteric conjugated with metasurface) the *Q*-factor of the COTS is ≈26, as experimentally demonstrated in Reference [28].

The spectral manifestation of the COTS can be described using the time-dependent coupled mode theory (TCMT) [36,37]. In the TCMT, a localized mode has eigenfrequency $\omega_{0}$ and energy relaxation times $\tau_{l}$, l=1…N in *N* channels through which energy can flow into or out of a localized mode. In the initial CLC–PPAM structure, the energy stored in the COTS can escape through three channels. The first channel with relaxation time $\tau_{\mathrm{PPAM}}^{20}$ is related to the transmission of light through the final PPAM. The superscript 20 at $\tau_{\mathrm{PPAM}}$ indicates the number of PPAM periods. The second channel with relaxation time $\tau_{\varphi_{1}}$ is related to the polarization loss at the PPAM–CLC interface. The third channel with relaxation time $\tau_{L}$ is related to the transmittance of the cholesteric layer of finite thickness. In this case, the spectral resonance linewidth corresponding to the COTS is determined by a total rate of the energy leakage from the localized mode and can be determined as [33]
(1)$$\frac{1}{\tau }=\frac{1}{\tau_{\mathrm{PPAM}}^{20}}+\frac{1}{\tau_{\varphi_{1}}}+\frac{1}{\tau_{L}}.$$

In addition, as it was shown in Reference [33], the third term in Equation (Equation 1) is negligible in comparison with the first two terms, since, at the chosen parameters, the rate of energy leakage through the CLC appears to be two orders of magnitude lower as compared with the leakage through the PPAM and the polarization loss at the boundary. It can be seen in Figure 1a that the localized state is close to the critical coupling condition, in which the amplitude of the corresponding resonance dip in the reflectance spectrum virtually vanishes [38]. This condition implies the equality of the rates of energy inflow into and outflow from the localized mode: $1/\tau_{\varphi_{1}}\approx 1/\tau_{\mathrm{PPAM}}^{20}$. In terms of the corresponding inverted *Q*-factors (damping ratios), Equation (Equation 1) can be presented in the form:(2)$$\frac{1}{Q}\approx \frac{1}{Q_{\mathrm{PPAM}}^{20}}+\frac{1}{Q_{\varphi_{1}}}.$$

For the COTS shown in Figure 1a, we have $1/Q\approx 7\times 10^{-3}$. Consequently, $1/Q_{\mathrm{PPAM}}^{20}\approx 1/Q_{\varphi_{1}}\approx 3.5\times 10^{-3}$. When the MS is embedded in the structure, Equation (Equation 1) acquires additional terms, since the leakage through the PPAM transmission channel is replaced by a sum of the polarization leakage at the PPAM–MS interface and the leakage into the MS absorption channel:(3)$$\frac{1}{Q_{\mathrm{PPAM}}^{20}}=\frac{1}{Q_{\varphi_{2}}}+\frac{1}{Q_{\mathrm{Abs}}},$$
where $\varphi_{2}$ is the angle between optical axes of the PPAM and MS.

The leakage into the MS absorption channel can be determined using the numerical calculation. As was shown in Reference [39], equating the imaginary part of the permittivity of a metal to zero leads to an increase in the *Q*-factor, since, in this case, the resonance linewidth is determined exclusively by the radiation loss. The FDTD calculation shows that the *Q*-factor of the localized state is $1/Q\approx 3.8\times 10^{-3}$ (please see Figure 2b). However, if Au losses are artificially eliminated, ($ℑ(\varepsilon_{\mathrm{Au}})=0$), the decay rate of the localized state is $1/Q_{\mathrm{Abs}=0}\approx 2\times 10^{-3}$. The absorption contribution to the resonant linewidth can be determined by subtraction: $1/Q_{\mathrm{Abs}}=1/Q-1/Q_{\mathrm{Abs}=0}\approx 3.8\times 10^{-3}-2\times 10^{-3}=1.8\times 10^{-3}$. At the chosen parameters, the leakage into the MS absorption channel was found to be comparable to the polarization leakage at the PPAM–MS interfaces.

The golden parts of the metasurface are the only considerable origin of material loss and material dispersion [35]. Here, we can neglect losses and dispersion in the other materials as they do not lead to considerable change in *Q*-factors. The all-dielectric layers of PPAM are assumed to be lossless and dispersionless, as well as the CLC layer.

The wavelength of the COTS inside the band gap is determined by the phase matching condition, which can be written in the form [33]:(4)$$\pi N=\theta +\chi +\varphi_{1}.$$ Here, θ and χ are the phases of the waves reflected from the PPAM and CLC, respectively. Similarly, we can describe the low-*Q* resonance at the PPAM-MS interface. Such a resonance for the case of the MS simplified to a uniform metallic layer was described in Reference [40]:(5)$$\pi N=\theta +\chi_{2}+\varphi_{2}.$$ Here, θ and $\chi_{2}$ are the phases of the waves reflected from the PPAM and CLC, respectively. The geometric phase incursion for a cycle of two re-reflections is $2\varphi_{2}$. The reflectance spectra of the structure at different angles $\varphi_{2}$ are presented in Figure 4.

It can be seen that the rotation of nanobricks can ensure a shift of the COTS and low-*Q* resonance on the MS to both the short- and long-wavelength spectral regions.

The proposed structure can be used in lasing. A laser model with the use of the Tamm modes at the interface between two periodic dielectric structures was discussed in Reference [14]. In Reference [41], it was first reported on the implementation of single-mode lasing using TPPs, which appear at the interface between a multilayer dielectric structure and a metal. For this purpose, quantum dots were placed on a Bragg reflector surface that was conjugated with a silver film. It should be noted that the TPPs are promising for use in electrically pumped microlasers, since the metallic layer can serve both as a mirror and as an electrode. A liquid-crystal laser was analytically discussed in References [42,43]. The photonic properties and possible lasing of one-dimensional periodic anisotropic structures with the double helix symmetry were reviewed in Reference [44]. A CLC-based laser was demonstrated in Reference [45] via embedding a dye.

In this work, we propose a model of a COTS-based vertical laser, in which a CLC serves as an active layer. The active medium was modeled by introducing the negative imaginary part *k* of the refractive index of a dye-doped cholesteric. The results of the calculation are presented in Figure 5. Interestingly, at small values (around k=−0.003) of the imaginary part of the cholesteric refractive index, the reflection from the structure is minimum or, in other words, the condition of a critical coupling of the incident field with the COTS is fulfilled. In this case, the total rate of energy leakage from the localized mode takes the form:(6)$$\frac{1}{\tau }=\frac{1}{\tau_{\varphi_{1}}}+\frac{1}{\tau_{\varphi_{2}}}+\frac{1}{\tau_{\mathrm{Abs}}}-\frac{1}{|\tau_{L}|}.$$
where $1/|\tau_{L}|$ is the rate of energy inflow into the localized mode caused by the negative *k* value. This term reduces to the sum of the energy leakage rates $1/\tau_{\varphi_{2}}+1/\tau_{\mathrm{Abs}}$, equating the total rate of energy leakage to the inflow rate $1/\tau_{\varphi_{1}}$ [33]. A further increase in the imaginary part leads to the sharp growth of the reflectivity at k<−0.009. Similar results can be obtained by doping the PPAM, rather than the CLC; then, the lasing threshold is overcome at k<−0.008. In a true experiment, the lasing efficiency is determined by the intensity of photoluminescence of the dye diluted in the medium. In the proposed structure this intensity should be maximal near the chiral optical Tamm state wavelength. Such a potential photonic device to experimentally demonstrate a COTS-based vertical laser can be based on a dye-embedded metasurface [46] or a liquid crystal-based laser [45].

## 4. Conclusions

In this study, the possibility of forming a chiral optical Tamm state at the interface between a cholesteric liquid crystal and a polarization-preserving anisotropic mirror conjugated to a metasurface was demonstrated. To reduce the loss, the metal part of the mirror is separated from the interface as far as possible. The advantage of the proposed model is that the *Q*-factor of the chiral optical Tamm state remains comparable to that of the localized state in an all-dielectric structure. The introduction of a metasurface into the structure makes it possible to decrease the number of PPAM layers from 20 to 8. In addition to miniaturization, this approach noticeably facilitates fabrication of the structure by providing better alignment and flatness or parallelism of interfaces. A drawback of introducing the metasurface is a possible low-*Q* resonance; this resonance, however, can be effectively suppressed by considerable inclination of the metasurface nanobricks relative to the optical axis of the multilayer PPAM. Thus, in the proposed structure, the chiral optical Tamm state wavelength is sensitive not only to the angle between the CLC and PPAM optical axes but also to the angle between the PPAM and metasurface ones. It was shown that the proposed structure can act as a chiral vertically emitting laser, and a cholesteric agent can serve as an active medium. This cholesteric layer can be simultaneously used for tunability and sensing by applying temperature or voltage.

## Figures and Tables

**Figure 1 nanomaterials-12-00234-f001:**
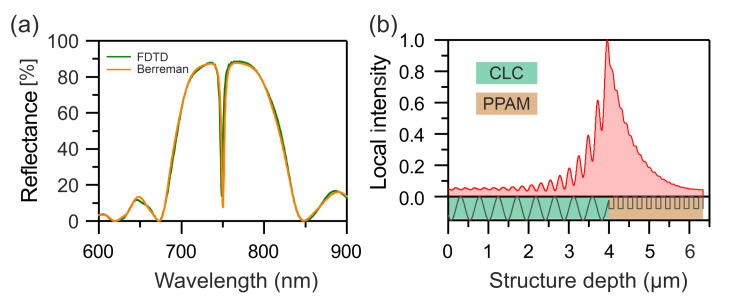
(**a**) Reflection spectra of the CLC–PPAM structure calculated by the Berreman (orange line) and FDTD (green line) methods. (**b**) Local field intensity distribution in the CLC–PPAM structure at the COTS wavelength.

**Figure 2 nanomaterials-12-00234-f002:**
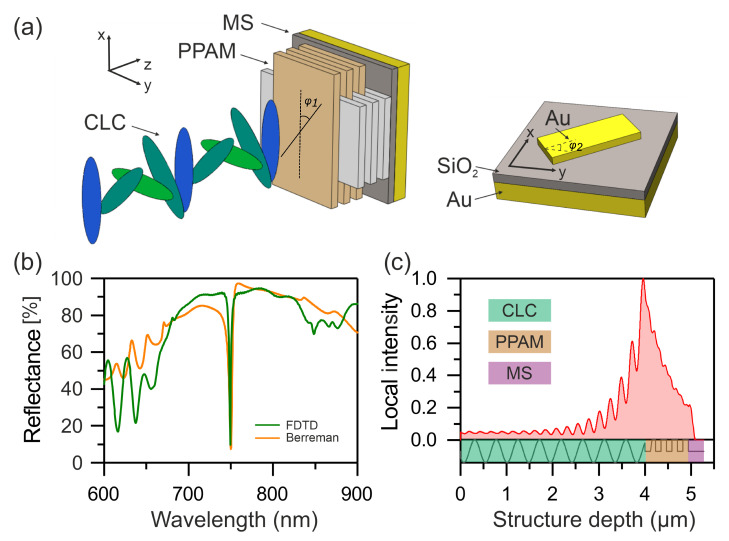
(**a**) Model of a hybrid structure for obtaining a high-*Q* COTS consisting of a CLC, PPAM, and MS (left). Schematic of the MS (right). $\varphi_{1}$ and $\varphi_{2}$ are the angles between the optical axes at the CLC-PPAM and PPAM-MS interfaces, respectively. (**b**) Reflectance spectra of the CLC–PPAM–MS structure calculated by the Berreman (orange line) and FDTD (green line) methods. (**c**) Local field intensity distribution at the COTS wavelength.

**Figure 3 nanomaterials-12-00234-f003:**
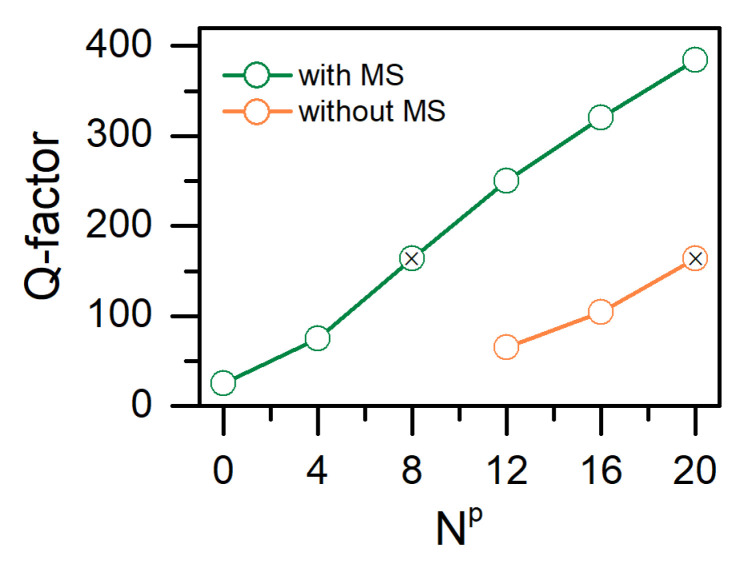
The dependence of the *Q*-factor of the chiral optical Tamm state on the number of PPAM periods, $N^{p}$, with and without the metasurface. The leftmost green open circle corresponds to the previous experimental result [28]. Black crosses have comparable *Q*-factors and correspond to Figure 2b.

**Figure 4 nanomaterials-12-00234-f004:**
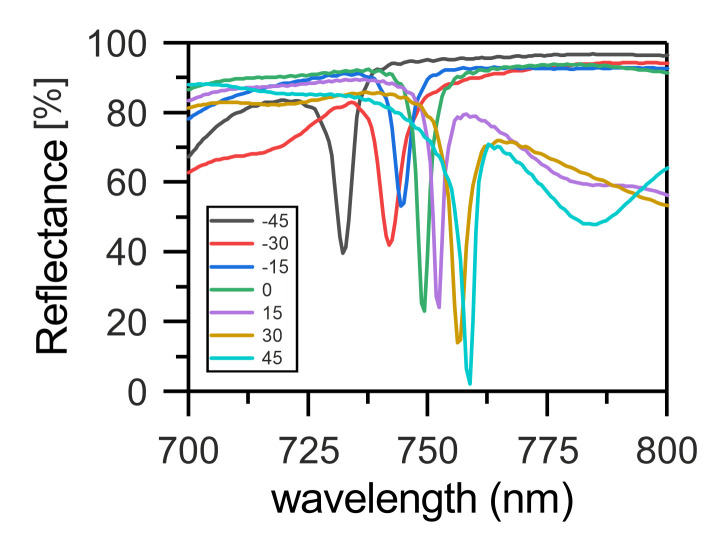
Reflectance spectra of the structure at different angles $\phi_{2}$. Positive angles correspond to the clockwise rotation of a nanobrick around the *z* axis, and negative angles correspond to the counterclockwise rotation.

**Figure 5 nanomaterials-12-00234-f005:**
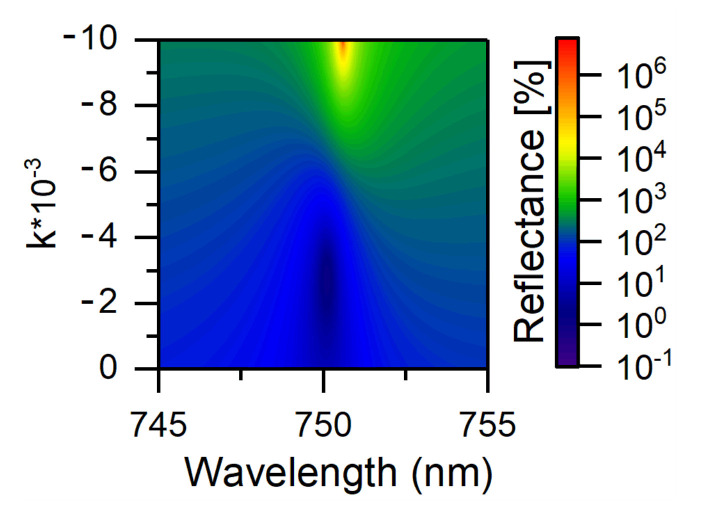
Reflectance spectra of the structure at negative values of the imaginary part *k* of the CLC refractive index. In contrast to passive structures, the reflectance goes beyond unity, as it is not limited by the energy conservation law. The sharp growth of the reflectivity at k<−0.009 indicates the lasing threshold [43].

## Data Availability

The data presented in this study are available upon reasonable request from the corresponding author.

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
