# Peer review of "Metal–Dielectric Polarization-Preserving Anisotropic Mirror for Chiral Optical Tamm State"

_nanomaterials, 2022, doi:10.3390/nano12020234_

Round 1

Reviewer 1 Report

The author demonstrated a chiral optical Tamm state localized at the interface between a cholesteric liquid crystal and a polarization-preserving anisotropic mirror. The suggested results are certainly helpful for the potential readers in the related area. However, it is concerned that all of results in the paper came from numerical calculation. And it is also same in their previous works. Therefore, I do not recommend this paper in the Nanomaterials.

1. It is not clear what is different between their old works Ref. 29 and 30. And the sentence "However, such a structure is difficult to be implemented." is not enough.
2. All of results in the paper came from numerical calculation. Please make real device and show its data.
3. The author showed only the calculated results. It should be clearfy in the abstract or title.
4. Please show the optical properties of the metasurface they used.
5. The topics in introduction is too narrow from the begining.
6. All the data ( reflectance/transmittance/absorptance) are shown without units.
7. Generally, liquid crystal based devices can be switched by applying an electric field. Can this device also be switchable?

Author Response

The author demonstrated a chiral optical Tamm state localized at the interface between a cholesteric liquid crystal and a polarization-preserving anisotropic mirror. The suggested results are certainly helpful for the potential readers in the related area. However, it is concerned that all of results in the paper came from numerical calculation. And it is also same in their previous works. Therefore, I do not recommend this paper in the Nanomaterials.

Comment 1. It is not clear what is different between their old works Ref. 29 and 30. And the sentence "However, such a structure is difficult to be implemented." is not enough.

Response: We thank the Reviewer for constructive comment that greatly motivates us for the research.

In our previous work [29. Rudakova, N.V.; Timofeev, I.V.; Vetrov, S.Y.; Lee, W. All-dielectric polarization-preserving anisotropic mirror. OSA Continuum 2018, 1, 682. doi:10.1364/osac.1.000682.] we explore a layered structure consisting of alternating uniaxial dielectric layers with orthogonally directed optical axes, which can be used to obtain the chiral optical Tamm state at the interface with the cholesteric liquid crystal layer. Later in the following work [30. Rudakova, N.V.; Timofeev, I.V.; Bikbaev, R.G.; Pyatnov, M.V.; Vetrov, S.Y.; Lee, W. Chiral Optical Tamm States at the Interface between an All-Dielectric Polarization-Preserving Anisotropic Mirror and a Cholesteric Liquid Crystal. Crystals 2019, 9, 502. doi:10.3390/cryst9100502.] we highlight the high Q-factor chiral optical Tamm state which exists at the interface between a cholesteric liquid crystal and a polarization-preserving anisotropic mirror consisting of 30 layers. However, for obtaining a multilayer structure, which should be strictly parallel, the technology has not yet been developed. In the current work, we have shown that a structure consisting of only 8 dielectric anisotropic layers can show high Q-factor when adding a metasurface. This metasurface-assisted structure has already been studied experimentally [Lin, M.Y.; Xu, W.H.; Bikbaev, R.G.; Yang, J.H.; Li, C.R.; Timofeev, I.V.; Lee, W.; Chen, K.P. Chiral-Selective Tamm Plasmon Polaritons. Materials 2021, 14. doi:10.3390/ma14112788.].

We have amended the crucial place in the Introduction section to comply with the Reviewer’s opinion:

However, such a structure is difficult to be implemented. In accordance with current technology requirements, one has to reduce the number of anisotropic layers until it reaches the experimentally available level described in \cite{Lin2021meta}.

Comment 2. All of results in the paper came from numerical calculation. Please make a real device and show its data.

Response: The Reviewer is right that all the results in the paper came from numerical calculation. We do our best to reach the real device level. Our current experiments on the chiral and LC-assisted tunable optical Tamm states have been published recently [Lin, M.Y.; Xu, W.H.; Bikbaev, R.G.; Yang, J.H.; Li, C.R.; Timofeev, I.V.; Lee, W.; Chen, K.P. Chiral-Selective Tamm Plasmon Polaritons. Materials 2021, 14. doi:10.3390/ma14112788;  Pankin P.S. et al. Experimental implementation of tunable hybrid Tamm-microcavity modes // Appl. Phys. Lett. 2021. Т. 119. â„– 16. С. 161107.]. This work is aimed at optimizing the Q-factor of such localized states. Indeed, we hope to reach experimental validation in the near future with the help of this theoretical publication.

The conclusion is improved to stress this fabrication intention:

In addition to miniaturization, this approach noticeably facilitates fabrication of the structure by providing better alignment and flatness or parallelism of interfaces.

Comment 3. The author showed only the calculated results. It should be clearfy in the abstract or title.

Response: We thank the Reviewer for this comment. Note was added to abstract:

«This numerical study demonstrates the possibility of exciting a chiral optical Tamm state localized at the interface between a cholesteric liquid crystal and a polarization-preserving anisotropic mirror conjugated to a metasurface.»

Comment 4. Please show the optical properties of the metasurface they used.

Response: We thank the Reviewer for this good comment. The refractive index of the SiO2 layer and reference to the gold dispersion have been added to the text:

«The MS is an array of rectangular gold [35] nanobricks arranged on a 100-nm-thick SiO2 layer deposited onto a 200-nm-thick reflecting gold plate. The refractive index of SiO_2 thin layer was taken to be 1.45.»

The reflection spectra of the metasurface were investigated in our previous work [Lin, M.Y.; Xu, W.H.; Bikbaev, R.G.; Yang, J.H.; Li, C.R.; Timofeev, I.V.; Lee, W.; Chen, K.P. Chiral-Selective Tamm Plasmon Polaritons. Materials 2021, 14. doi:10.3390/ma14112788.]. Please see the replicate of Fig. 3 below. (Please see the attached file).

Comment 5. The topics in introduction is too narrow from the begining.

Response: The Reviewer is right that the topics in introduction were focusing on the optimization of Q-factor for localized states. This was made intentionally, not to expand the introduction to a large size. Now we have modified the Introduction section. For each item, appropriate links are provided for a broader overview. Please, see the changes in responses to comment 1 and third Reviewer’s comment 4.

Comment 6. All the data (reflectance/transmittance/absorptance) are shown without units.

Response: We thank the Reviewer for this comment. The units have been added.

Comment 7. Generally, liquid crystal based devices can be switched by applying an electric field. Can this device also be switchable?

Response: The Reviewer is right. The optical properties of the chiral optical Tamm state can be tuned by applying an electric field. The metasurface can serve as an electrical contact. However, it is a separate technical problem to make the structure more complicated by the second electrical contact. On the other hand, additional switching can be easier demonstrated by temperature. This is especially applicable for a liquid crystalline substance with a narrow mesophase temperature range. In our previous work, the influence of temperature on the chiral optical Tamm state wavelength was demonstrated. Please see the copy of Fig. 6 from [28]: (Please see the attached file).

Following the reviewer’s comment, the Conclusion section has been appropriately modified:

It was shown that the proposed structure can act as a chiral vertically emitting laser, and a cholesteric agent can serve as an active medium. This cholesteric layer can be simultaneously used for tunability and sensing by applying temperature or voltage.

Reviewer 2 Report

The paper is very interesting. They show the modes with infinity quality factor. The nunerical studies are exhaustive

 Therefore, the paper ke suitable for publication after the following issue.

1) they should be discuss the role of the loss 

2) the material proposed with permittivity of 24.5, the authors could show the reference and the eventual dispersion in frequency 

3) the authors should be show the reception

Author Response

The paper is very interesting. They show the modes with infinity quality factor. The numerical studies are exhaustive

 Therefore, the paper ke suitable for publication after the following issue.

We thank the Reviewer for the positive evaluation of our paper.

 Comment 1: they should be discuss the role of the loss

Response: We thank the reviewer for this comment. The metasurface material loss in gold nano-bricks and reflecting gold plate was accounted for both numerically [P. B. Johnson and R. W. Christy. Optical constants of the noble metals, Phys. Rev. B 6, 4370-4379 (1972)] and theoretically, see Q_{Abs} in eq. (3). In this research we can neglect any loss in the other materials as it does not change the Q-factors considerably. Such reasonable omission makes the model simpler and easier to reproduce by the others in the field. Corresponding discussion was inserted to the manuscript:

The golden parts of the metasurface are the only considerable origin of material loss and material dispersion [P. B. Johnson and R. W. Christy. Optical constants of the noble metals, Phys. Rev. B 6, 4370-4379 (1972)]. Here we can neglect losses and dispersion in the other materials as they do not lead to considerable change in Q-factors. The all-dielectric layers of PPAM are assumed to be lossless and dispersionless, as well as the CLC layer.

Comment 2: the material proposed with permittivity of 24.5, the authors could show the reference and the eventual dispersion in frequency

Response: We thank the reviewer for this comment. The all-dielectric parts are considered dispersionless, the metallic part dispersion is conventionally accounted for [P. B. Johnson and R. W. Christy. Optical constants of the noble metals, Phys. Rev. B 6, 4370-4379 (1972)]. Corresponding discussion was inserted to the manuscript and presented in the response to the Reviewer’s comment 2.

Comment 3:  the authors should be show the reception

Response: Fabrication of the metasurface and CLC-metasurface devices was covered in detail in the Supplementary Information of our previous article. The reception is bulky, so we advise the Reviewer to take a look at the details by following the link: https://www.mdpi.com/1996-1944/14/11/2788#supplementary

In the manuscript we refer this work as [28. Lin, M.Y.; Xu, W.H.; Bikbaev, R.G.; Yang, J.H.; Li, C.R.; Timofeev, I.V.; Lee, W.; Chen, K.P. Chiral-Selective Tamm Plasmon Polaritons. Materials 2021, 14].

Reviewer 3 Report

Aiming at the generation structure of chiral optical Tamm state, the authors have added a metasurface below the polarization-preserving anisotropic mirror (PPAM) multilayers, to reduce the number of layers needed in PPAM, simplify the fabrication and makes it possible to control the spectral properties. Here are my comments and questions:

  1. In figure2 (b), what causes the difference between FDTD and Berreman calculated reflectance spectrum? How did the shape of metasurface get considered in the Berreman method? Can the two spectrums get closer by refining the FDTD mesh?

  1. The number of PPAM layers had decreased from 20 to 8 with the help of the bottom metasurface in the example, but the influence of layer numbers and metasurface is still not clear. I suggest the authors add two curves of Q-factors varied with layer numbers, under conditions of with and without the bottom metasurface.

  1. This work lacks the verification of experimental results, is it possible to have some experimental results to verify the effects of metasurface?

  1. Some metasurface based polarization-preserving anisotropic mirrors have been reported elsewhere, e.g., ACS Photonics 3(11): 2096–2101, (2016), Scientific Reports 5: 8434 (2015), Opto-Electron Adv 4, 200024 (2021), which may be helpful to enrich the introduction.

Author Response

Aiming at the generation structure of chiral optical Tamm state, the authors have added a metasurface below the polarization-preserving anisotropic mirror (PPAM) multilayers, to reduce the number of layers needed in PPAM, simplify the fabrication and makes it possible to control the spectral properties. Here are my comments and questions:

Comment 1: In figure2 (b), what causes the difference between FDTD and Berreman calculated reflectance spectrum? How did the shape of metasurface get considered in the Berreman method? Can the two spectrums get closer by refining the FDTD mesh?

Response: We thank the Reviewer for this comment. The difference between FDTD and Berreman reflectance spectra in Figure 2b is explained by the simplifications implemented in the Berreman method. In this case, the nanobrick layer was presented as an anisotropic homogeneous layer in x-y directions, while the FDTD method does not need such simplifications. Thus, increasing the mesh will not lead to better convergence between the results derived from the two methods. Corresponding explanations have been added to the text.

«In order to fulfil the one-dimensional assumption for the Berreman method, the subwavelength nanobrick array was reasonably considered homogeneous in x-y directions at the wavelength scale. It was presented as an anisotropic layer with refractive indices noms = 0.5 and nems = 5.5, respectively.»

Comment 2: The number of PPAM layers had decreased from 20 to 8 with the help of the bottom metasurface in the example, but the influence of layer numbers and metasurface is still not clear. I suggest the authors add two curves of Q-factors varied with layer numbers, under conditions of with and without the bottom metasurface.

Response: The Reviewer is right, whose valuable comment has been very significant to help improve the quality of our work. The new figure with Q-factor of the chiral optical Tamm state has been added to the text.

«It is important to note that, for PPAM with the number of layers less than 12 without metasurface, a COTS is not spectrally resolved. Moreover, the attached metasurface provides the excitation of a localized state even with fewer layers of PPAM. Thus, when Np = 0 (cholesteric conjugated with metasurface) the Q-factor of the COTS is ~26, as experimentally demonstrated in [28].

Please see the attached file to view Figure 3.

Figure 3. The dependence of the Q-factor of the chiral optical Tamm state on the number of PPAM periods, Np, with and without the metasurface. The leftmost green open circle corresponds to the experimental result [28]. Black crosses have comparable Q-factors and correspond to Fig. 2b»

Comment 3: This work lacks the verification of experimental results, is it possible to have some experimental results to verify the effects of metasurface?

Response: We thank the Reviewer for this comment. The COTS at the CLC-metasurface interface has been investigated experimentally in our previous work [Lin, M.Y.; Xu, W.H.; Bikbaev, R.G.; Yang, J.H.; Li, C.R.; Timofeev, I.V.; Lee, W.; Chen, K.P. Chiral-Selective Tamm Plasmon Polaritons. Materials 2021, 14. doi:10.3390/ma14112788.]. Experimental validation of the presented model is planned to be carried out in the future.

Comment 4: Some metasurface based polarization-preserving anisotropic mirrors have been reported elsewhere, e.g., ACS Photonics 3(11): 2096–2101, (2016), Scientific Reports 5: 8434 (2015), Opto-Electron Adv 4, 200024 (2021), which may be helpful to enrich the introduction.

Response: We thank the Reviewer for references. The references have been added to the Introduction section. In the list of references numbers 29, 30 and 31.

«The second way is to combine a chiral medium with an anisotropic mirror [23,28–31].»
